# COVIDLIES: Detecting COVID-19 Misinformation on Social Media

**Tamanna Hossain**[*,◇]    **Robert L. Logan IV**[*,◇]    **Arjuna Ugarte**[*,♠]    **Yoshitomo Matsubara**[*,◇]
**Sean Young**[♠]    **Sameer Singh**[◇]
◇ Dept. of Computer Science, University of Califonia, Irvine
♠ Dept. of Emergency Medicine, University of Califonia, Irvine
{tthossai, rlogan, dugarte, yoshitom, syoung5, sameer}@uci.edu

## Abstract

The ongoing pandemic has heightened the need for developing tools to flag COVID-19-related misinformation on the internet, specifically on social media such as Twitter. However, due to novel language and the rapid change of information, existing misinformation detection datasets are not effective for evaluating systems designed to detect misinformation on this topic. Misinformation detection can be divided into two sub-tasks: (i) retrieval of misconceptions relevant to posts being checked for veracity, and (ii) stance detection to identify whether the posts Agree, Disagree, or express No Stance towards the retrieved misconceptions. To facilitate research on this task, we release COVIDLIES[1], a dataset of 6761 expert-annotated tweets to evaluate the performance of misinformation detection systems on 86 different pieces of COVID-19 related misinformation. We evaluate existing NLP systems on this dataset, providing initial benchmarks and identifying key challenges for future models to improve upon.

## 1 Introduction

Detecting spread of misinformation such as, rumors, hoaxes, fake news, propaganda, spear phishing, and conspiracy theories, is an important task for natural language processing (Thorne et al., 2017; Shu et al., 2017; Thorne and Vlachos, 2018). Online social media networks provide particularly fertile ground for the spread of misinformation—they lack gate-keeping and regulations, users publish content without having to go through an editor, peer review, verification of qualification, or providing sources, and social networks tend to create "echo chambers" or closed networks of communication insulated from disagreements.

---

[*]First four authors contributed equally.
[1]https://ucinlp.github.io/covid19

---

**Tweet:** "Coronavirus CV19 was a top secret biological warfare experiment. That is why it is only affecting the poor."
**Misconception:** "Coronavirus is genetically engineered."
**Label:** Agree

---

**Tweet:** "It looks like we are all going to have to wait much longer for a #COVID19 vaccine."
**Misconception:** "We're very close to a vaccine."
**Label:** Disagree

---

**Tweet:** "CDC: Coronavirus spreads rapidly in dense populations with public transit and regular social gatherings."
**Misconception:** "Coronavirus cannot live in warm and tropical temperatures."
**Label:** No Stance

---

Figure 1: **COVIDLIES Dataset.** Given a *tweet*, we annotate whether any of the known *misconceptions* are expressed in the tweet, in particular, if the tweet spreads the misconception (e.g., they Agree), combats the spread of the misconception (e.g., they Disagree), or takes No Stance towards the misconception.

The COVID-19 pandemic has created a pressing need for tools to combat the spread of misinformation. Since the pandemic affects the global community, there is a wide audience seeking information about the topic, whose safety is threatened by adversarial agents invested in spreading misinformation for political and economic reasons. Furthermore, due to the complexity of medical and public health issues, it is also difficult to be completely accurate and factual, leading to disagreements that get exacerbated with misinformation. This difficulty is compounded by the rapid evolution of knowledge regarding the disease. As researchers learn more about the virus, statements that seemed true may turn out to be false, and vice versa. Detecting this spread of pandemic-related misinformation, thus, has become a critical problem, receiving significant attention from government and public health organizations (WHO, 2020), social media platforms (TechCrunch, 2020), and news agencies (BBC, 2020; CNN, 2020; New York Times, 2020).

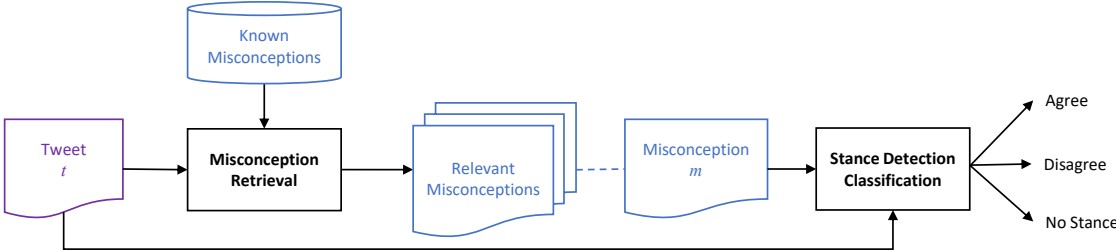

Figure 2: **Misconception Detection Pipeline** consisting of two sub-tasks, (a) *Misconception Retrieval* that identifies the known misconceptions that are relevant to the given tweet, and (b) *Stance Detection* that identifies whether the tweet agrees, disagrees, or expresses no stance, for each of the relevant misconceptions.

In this paper, we introduce the COVIDLIES dataset for misconception detection on Twitter. COVIDLIES comprises of 86 common misconceptions about COVID-19 along with 6761 related tweets, identified and annotated by researchers from the UCI School of Medicine. Given a tweet, we annotate whether any of the *known* misconceptions, curated by the researchers, are expressed by the tweet. If they are not, then they are considered No Stance. If they are, we further identify whether the tweet propagates the misconception (Agree) or is informative by contradicting it (Disagree). Example misconception-tweet pairs for each label are illustrated in Figure 1.

We provide benchmark results for existing NLP models for this task. First, we evaluate text similarity models on their ability to detect whether a tweet is relevant to a given misconception (a.k.a *misconception retrieval*). Following prior work on fact verification (Thorne et al., 2018) and fake news detection (Yang et al., 2019), we evaluate NLI models on misinformation (a.k.a. *stance detection*), by equating the class labels Agree, Disagree, and No Stance to Entailment, Contradiction, and Neutral, respectively. Our results show that existing models struggle at both tasks (38.7 Hits@1 for retrieval and 32.5 macro F1 on stance detection), however improve considerably after domain adaptation (Gururangan et al. (2020); 61.3 Hits@1 for retrieval and 50.2 macro F1 on stance detection).

While our initial results using domain adaptation are encouraging, they leave much room for improvement. There is still much work that needs to be done before NLP systems can be seriously considered for combating COVID-19-related misinformation, and we hope COVIDLIES will be useful to help researchers understand when such systems are ready to be deployed.

## 2   Problem Setup

We assume access to a collection of positively phrased *known* misconceptions $M = \{m_1, \ldots, m_{|M|}\}$, e.g., "Wearing masks does not prevent spread of COVID-19." is a misconception. As we describe later, the set of misconceptions in this work are vetted, curated, and maintained by medical researchers. Given a collection of tweets, $T = \{t_1, \ldots, t_{|T|}\}$, the task is to determine, for each input $t$, whether there exists a misconception $m \in M$ that is being discussed, and if so, whether the discussion propagates the misconception (i.e., identifies $m$ as true, and thus is spreading the misconception) or refutes the misconception (i.e., identifies $m$ as false). This task is naturally separated into the following steps (shown in Figure 2):

1. **Misconception Retrieval:** Given $t$ return a subset $M_t \subseteq M$ of relevant misconceptions.
2. **Stance Detection:** For each $(m, t)$ pair ($m \in M_t$), predict whether the $m$ and $t$ Agree, Disagree, or $t$ takes No Stance with respect to $m$.

Due to limited availability of labeled data specific to this problem, we expect that models will need to be supervised on other, related tasks. For misconception retrieval, for example, relevant misconceptions can be ranked by measuring the semantic similarity between the tweet and each misconception, e.g., using cosine similarity between average word embeddings or more recent transformer-based methods such as BERTSCORE (Zhang et al., 2019). For the stance detection sub-task, the problem can be recast as natural language inference (NLI), mapping the tweet $t$ to the premise, the misconception $m$ to the hypothesis, and the Agree, Disagree, and No Stance labels to Entailment, Contradiction, and Neutral, respectively.

## 3 Dataset Collection

Due to novel language used to describe the disease and its associated misconceptions, existing misinformation detection dataset are unlikely to be effective for evaluating systems designed to detect COVID-19-related misinformation on social media. To facilitate research on this problem, we collect an evaluation dataset, COVIDLIES; the collection process is described below.

**Misconceptions** We extract misconceptions from a Wikipedia article about misinformation related to the COVID-19 pandemic (Wikipedia, 2020). The extracted statements are manually examined, and statements that are not misinformation are removed. We manually rephrase the misinformation statements to a positive expression of that misinformation, e.g. "Some conspiracy theorists also alleged that the coronavirus outbreak was cover-up for a 5G-related illness" is shortened to "Coronavirus is caused by 5G".

**Tweets** Our source of tweets is the collection of COVID-19-related tweets identified by Chen et al. (2020). We only use tweets from March and April 2020, and filter out non-English tweets.

**Annotation Process** To help identify tweets related to our list of misconceptions, we use BERTSCORE (Zhang et al., 2019) to compute a similarity metric on tweet-misconception pairs. For each given misconception, the 100 most similar tweets are selected for annotation. Each of these tweet-misconception pairs is manually labeled by researchers in the UCI School of Medicine as either: Agree (tweet is a positive expression of the misconception), Disagree (tweet contradicts/disagrees with the misconception), or No Stance (tweet is neutral or not relevant to the misconception).

**Annotation Quality** To evaluate inter-rater reliability, we randomly chose a subset of 200 tweet-misconception pairs and had four researchers manually label the subset. Percent agreement between researchers was 79%. Fleiss Kappa score was 0.69 which indicates substantial agreement between researchers (0.61–0.8). Disagreements were discussed and resolved before continuing to label the remaining tweet-misconception pairs in the dataset.

Most disagreements came down to labeler interpretation. For example, given the misconception, "Drinking large amounts of water will protect against coronavirus", and a tweet of "It's a good thing everyone is stocking up on water to survive the Coronavirus because the 128 OZ of Diet Coke, the double cheeseburgers, and radiation from our phones definitely won't kill us first", one researcher labeled the pair as No Stance because it does not address any protective benefits but another researcher labeled the pair as Agree because people were stocking up on water to survive Coronavirus, which, to that researcher, implied water was protective. After discussions among the researchers, we concluded the pair was No Stance as it did not implicitly address the benefits of water and the statement was stated in a sarcastic tone. Other labeling challenges included deciding whether or not links or images in the tweet should be taken into account, as these could potentially change context of the tweet. We concluded that we would only evaluate the text as is since the various models would not be able to take images and links into account.

**Dataset Statistics** The current dataset contains 86 misconceptions, along with 6761 annotated tweet-misconception pairs. Statistics about the distribution of labels are provided in Table 2. The distribution is heavily skewed, containing mostly No Stance tweets, and a higher proportion of Agree tweets than Disagree. The heavy skew towards No Stance tweets could be a due to the dataset construction methodology, specifically using BERTSCORE without fine-tuning to retrieve tweets per misconception. As we show in 4.2, domain adaptation significantly improves misconception matching. Further, presence of more Agree than Disagree tweets could be due to a bias in BERTSCORE towards scoring agreement higher.

Top misconceptions for each class are shown in Table 1. We only consider misconceptions with more than 80 annotated tweets, and rank the misconceptions for each class by the proportion of tweets that are annotated as that class. We present the top three misconceptions for each class with their corresponding percentage. There are misconceptions for which 100% of the paired annotated tweets express No Stance, which we do not see for the other two classes. We also notice that there are misconceptions with greater than 60% of paired tweets labeled as Agree; however, the highest proportion of Disagree labeled tweets found for any misconception in the Disagree class was 51%.

COVIDLIES, however, is an evolving dataset; annotation is not yet complete for all 86 Wikipedia misconceptions matched to 100 tweets using

| Class | Misconception | % |
|---|---|---|
| Agree | Democrats are using the coronavirus situation to harm President Trump. | 65.0 |
| | Coronavirus was taken from a Canadian lab or is the result of bioweapons defense research in China. | 60.2 |
| | The media is intentionally stoking fears of COVID-19 to destabilize the Trump administration. | 56.3 |
| Disagree | COVID-19 is only as deadly as the seasonal flu. | 51.0 |
| | Anybody in the U.S. who wants a COVID-19 test can get a test. | 36.7 |
| | The U.S. containment of the virus is 'close to airtight'. | 35.4 |
| No Stance | Acetic acid is effective against coronavirus. | 100.0 |
| | Cannabis protects against COVID-19. | 100.0 |
| | Clapping will kill coronavirus. | 100.0 |

Table 1: **Top Misconceptions by Class.** Misconceptions with more than 80 tweets total are ranked by the percentage of tweets annotated for each class. The top three misconceptions for each class with the corresponding percentage that a paired tweet would be annotated as that respective class are shown. For example, for the misconception 'Democrats are using the coronavirus situation to harm President Trump', 65% of the tweets paired with this misinformation were annotated as Agree.

| Class | Count | Percentage |
|---|---|---|
| Agree | 670 | 9.91 % |
| Disagree | 343 | 5.07 % |
| No Stance | 5,748 | 85.02 % |

Table 2: **Distribution of Labels** in the annotations.

BERTSCORE, and we are continually identifying additional misconceptions, as well as collecting more recent tweets for annotation. Further, we will gather more relevant tweets by using domain-adapted retrieval models, which, as we will see in the next section, considerably outperform the current approach to retrieval, BERTSCORE.

## 4 Performance of Benchmark Models

Supervised classifiers have been used extensively for detecting misinformation (Wang, 2017; Karimi et al., 2018; Shu et al., 2017, 2019). However, existing tasks involve static or slowly evolving domains, and topics that do not require specific expertise to annotate. Gathering an annotated dataset large enough to be used for training a COVID-19 misinformation detector is difficult: the way misconceptions are expressed rapidly evolves, and identifying whether or not something is a misconception requires expertise in public health and medicine. Further, even the misconceptions themselves change over time as we learn more about the disease and the pandemic. Thus, it is desirable that COVID-19 misinformation detection systems are: (i) *data efficient*, e.g., trained with little to no supervision, and (ii) *flexible*, e.g., allow the addition, removal, or modification of the known misconceptions.

In this section, we investigate whether models trained for related tasks in natural language processing can be adapted to misinformation detection on the COVIDLIES dataset without additional training. We specifically focus on models that can be used to score two input sequences, i.e., tweet-misconception pairs. Because these models come pretrained on different tasks, they are naturally *data efficient*, and furthermore, due to their pairwise nature, are also *flexible* as modification of supported misconceptions is performed at the input level. Our code, dataset, and a demo of our best performing system are all available at https://ucinlp.github.io/covid19.

### 4.1 Evaluation Metrics

In the misconception retrieval sub-task, for a each tweet, $t$, the goal is to retrieve all the misconceptions that the tweet refers to (i.e. may be labeled Agree or Disagree by the annotators). Note: for clearer description, we introduce a new "pseudo-label", Relevant, to refer to misconceptions that either Agree or Disagree with a given tweet. We treat this as a ranking task, where for each tweet, $t$, the system ranks the list of misconceptions, $M$, in decreasing order of relevancy. We evaluate this ranking using the standard information retrieval metrics Hits@$k$ and Mean Reciprocal Rank (MRR) for each Relevant misconception $m \in M_t^*$.

The stance detection sub-task is a standard classification problem with three classes (Agree, Disagree, and No Stance). As such we perform evaluation by measuring the precision, recall, and F1-score of the predicted classes.

| Model | Agree | | | | Relevant (Agree or Disagree) | | | |
|---|---|---|---|---|---|---|---|---|
| | H@1 | H@5 | H@10 | MRR | H@1 | H@5 | H@10 | MRR |
| Cosine Sim., TF-IDF | 31.9 | 61.6 | 79.7 | 0.47 | 30.6 | 60.3 | 76.7 | 0.45 |
| BM25 | 38.7 | 68.2 | 77.2 | 0.52 | 37.3 | 68.2 | 78.6 | 0.51 |
| Cosine Sim., Avg. GloVe | 12.2 | 44.8 | 57.9 | 0.27 | 14.5 | 45.7 | 60.5 | 0.29 |
| Cosine Sim., Avg. BERT Embds. | 13.9 | 40.7 | 59.4 | 0.27 | 15.0 | 41.5 | 59.4 | 0.28 |
| BERTSCORE | 36.1 | 67.9 | 82.5 | 0.52 | 41.7 | 71.7 | 85.3 | 0.56 |
| *with Domain Adaptation (DA)* | | | | | | | | |
| Cosine Sim., Avg. BERT Embds. | 38.4 | 71.5 | 86.0 | 0.54 | 37.4 | 70.0 | 84.3 | 0.53 |
| BERTSCORE | **61.3** | **92.7** | **96.9** | **0.77** | **60.8** | **90.6** | **95.6** | **0.75** |

Table 3: **Misconception Retrieval Performance**. We present evaluation for misinformative tweets (e.g., tweets that Agree with one or more misconceptions), as well as combined evaluation on Relevant tweets (i.e., tweets that either Agree or Disagree with one or more misconceptions.)

| Model | Macro Avg. | | | Agree | | | Disagree | | | No Stance | | |
|---|---|---|---|---|---|---|---|---|---|---|---|---|
| | P | R | F1 | P | R | F1 | P | R | F1 | P | R | F1 |
| *Trained on SNLI* | | | | | | | | | | | | |
| Linear, Bag-of-Words | 33.1 | 35.4 | 27.7 | 8.2 | 16.1 | 10.9 | 6.6 | 42.6 | 11.5 | 84.5 | 47.6 | 60.9 |
| Linear, Avg. GloVe | 32.9 | 30.7 | 28.2 | 13.6 | 26.3 | 17.9 | 2.8 | 16.3 | 4.8 | 82.4 | 49.6 | 61.9 |
| BiLSTM | 33.2 | 36.4 | 27.4 | 8.9 | 15.8 | 11.4 | 6.7 | 47.8 | 11.7 | 84.1 | 45.7 | 59.2 |
| SBERT | 32.7 | 30.8 | 26.9 | 11.5 | 9.9 | 10.6 | 4.1 | 31.8 | 7.3 | 82.6 | 50.8 | 62.9 |
| SBERT (DA) | 33.8 | 30.4 | 22.7 | 22.2 | 11.9 | 15.5 | 4.1 | 46.9 | 7.5 | 75.2 | 32.3 | 45.1 |
| BERTSCORE (DA) + BiLSTM | 44.2 | 45.3 | 43.1 | 28.3 | 15.8 | 20.3 | 14.4 | 32.1 | 19.9 | 90.0 | 88.0 | 89.0 |
| BERTSCORE (DA) + SBERT (DA) | 49.3 | 44.4 | 42.6 | 46.7 | 14.6 | 22.3 | 11.3 | 30.6 | 16.5 | 90.0 | 88.0 | 89.0 |
| *Trained on MultiNLI* | | | | | | | | | | | | |
| Linear, Bag-of-Words | 35.2 | 38.1 | 24.0 | 9.8 | 59.7 | 16.9 | 10.5 | 28.9 | 15.4 | 85.3 | 25.8 | 39.7 |
| Linear, Avg. GloVe | 35.9 | 40.8 | 26.6 | 15.8 | 68.5 | 25.7 | 4.2 | 21.6 | 7.1 | 87.5 | 32.2 | 47.1 |
| BiLSTM | 32.0 | 33.6 | 32.5 | 10.8 | 6.4 | 8.1 | 0.0 | 0.0 | 0.0 | 85.1 | **94.2** | **89.5** |
| SBERT | 36.1 | 40.1 | 32.2 | 17.6 | 31.9 | 22.7 | 6.1 | 37.6 | 10.5 | 84.7 | 50.6 | 63.4 |
| SBERT (DA) | 51.1 | 47.3 | 41.5 | 58.1 | 23.4 | 33.4 | 8.7 | **50.4** | 14.9 | 86.5 | 67.9 | 76.1 |
| BERTSCORE (DA) + BiLSTM | 39.0 | 44.6 | 41.0 | 27.0 | 45.8 | 34.0 | 0.0 | 0.0 | 0.0 | 90.0 | 88.0 | 89.0 |
| BERTSCORE (DA) + SBERT (DA) | **55.9** | 50.9 | **50.2** | **63.3** | 30.6 | **41.2** | 14.4 | 34.1 | **20.3** | 90.0 | 88.0 | 89.0 |
| *Trained on MedNLI* | | | | | | | | | | | | |
| Linear, Bag-of-Words | 35.7 | 39.3 | 22.4 | 10.6 | 64.5 | 18.2 | 8.6 | 31.2 | 13.5 | 87.7 | 22.2 | 35.4 |
| Linear, Avg. GloVe | 39.9 | 50.3 | 28.2 | 13.4 | 74.8 | 22.8 | 12.3 | 48.7 | 19.6 | 94.1 | 27.3 | 42.3 |
| BiLSTM | 31.9 | 33.7 | 25.0 | 10.2 | 57.6 | 17.3 | 0.0 | 0.0 | 0.0 | 85.6 | 43.5 | 57.7 |
| SBERT | 35.9 | 37.0 | 16.3 | 10.5 | **87.6** | 18.8 | 8.9 | 12.2 | 10.3 | 88.1 | 11.1 | 19.7 |
| SBERT (DA) | 40.3 | **51.2** | 30.2 | 17.6 | 82.5 | 29.0 | 7.9 | 38.5 | 13.2 | **95.3** | 32.5 | 48.5 |
| BERTSCORE (DA) + BiLSTM | 43.7 | 44.6 | 41.5 | 27.0 | 44.9 | 33.7 | 14.3 | 0.9 | 1.6 | 90.0 | 88.0 | 89.0 |
| BERTSCORE (DA) + SBERT (DA) | 47.8 | 49.2 | 48.4 | 34.2 | 40.9 | 37.2 | **19.2** | 18.7 | 18.9 | 90.0 | 88.0 | 89.0 |

Table 4: **Stance Detection Performance.** We present evaluation for classification of tweet-misconception pairs into Agree, Disagree, and, No Stance classes. Precision (P), Recall (R), and F1-Score (F1) are presented for each class as well as macro averaged values. DA indicates domain-adaptive pretraining on COVID-19 tweets.

## 4.2 Misconception Retrieval

We evaluate a number of information retrieval and semantic similarity approaches for the misconception retrieval sub-task.

**Information Retrieval** We use two information retrieval approaches. The first approach uses TF-IDF vectorization of tweets and misconceptions. Cosine similarity is used to score each tweet-misconception pair. Misconceptions are retrieved for each tweet in decreasing order of this score. NLTK is used for tokenization and vectorization.

The second approach uses the BM25 algorithm, a bag-of-words retrieval technique which retrieves documents in decreasing probability of relevance of the query term. IDF and document lengths are used to determine probability of relevance. We use the pyserini implementation of BM25 to retrieve misconceptions for each tweet.

**Semantic Similarity** We obtain vectorized representations of tweets and misconceptions using word embeddings. We then use two approaches for computing the semantic similarity between them: (i) cosine similarity computed between average to-

ken embeddings, and (ii) BERTSCORE (Zhang et al., 2019), which involves computation over BERT token embeddings of the tweet and misconception to obtain an F1-score-like measurement that we use as a similarity score.

For the cosine similarity approach, we experiment with both non-contextualized and contextualized word embeddings. For non-contexualized word embeddings we use 300D GloVe trained on 2014-Wikipedia and Gigaword embeddings (Pennington et al., 2014). For contexualized embeddings we use a pretrained BERT-LARGE (Devlin et al., 2018) model. However, Since BERT is not trained on COVID-19-related text we also use COVID-Twitter-BERT[2] (Müller et al., 2020) which uses domain-adaptive pretraining (Gururangan et al., 2020) on 160M tweets about COVID-19. For sake of brevity, we will append the suffix (DA) to models that use COVID-Twitter-BERT instead of spelling out the full model name.

**Results** We present the performance of similarity models in Table 3. Average embedding, both with GloVe and BERT embeddings, perform the worst (and are fairly similar to each other). Although information retrieval based approaches, TF-IDF and BM25, considerably outperform the average embedding techniques, BERTSCORE captures the similarity as accurately as well. Domain adaptation, however, further improves the embedding-based similarity techniques, improving average BERT embeddings to be as good as others, while making BERTSCORE much more accurate than all other techniques. Thus we see that using domain adaptation and BERTSCORE are both important for performing accurate misconception retrieval.

We illustrate the differences in the similarity models using example predictions in Table 5. The first example provides a challenging case of retrieval that requires taking both COVID-19 knowledge and contextual information (e.g. multiple sentences, 'testing' vs 'tests') into account, and thus only the BERTSCORE (DA) model is able to retrieve the correct misconception. The second example primarily requires domain knowledge that 'coronavirus' and 'Sars-cov-2' are very similar, and only domain-adapted models are able to score the correct misconception highest. The last example shows when contextual embeddings (BERT) outperform non-contextual embedding (GloVe).

[2]https://huggingface.co/digitalepidemiologylab/covid-twitter-bert

### 4.3 Stance Detection, using NLI Models

Due to the lack of adequately large datasets for stance detection with pairs of sentences (Mohammad et al., 2016; Ferreira and Vlachos, 2016; Gorrell et al., 2018), we cannot use existing datasets to train models for our setup. However, since classes in misinformation detection correspond to those in natural language inference (NLI), a task with much larger training datasets, we instead experiment with adapting NLI models on this task.

We train linear classifiers on three common NLI datasets—SNLI (Bowman et al., 2015), MultiNLI (Williams et al., 2018), and MedNLI (Shivade, 2019). These classifiers use the following features, respectively: (i) concatenated unigram and bigram TF-IDF vectors for each input, (ii) concatenated average GloVe embeddings for each input, (iii) Bidirectional LSTM encoding, and (iv) the Sentence-BERT (SBERT) (Reimers and Gurevych, 2019) representation that uses siamese and triplet networks to obtain semantically meaningful sentence embeddings. Note that for (iii) and (iv), the transformer architectures (BiLSTM and SBERT) are jointly trained with the linear classifier.

**BERTSCORE (DA) + NLI** Since BERTSCORE with domain adaptation performs best at retrieval for relevant classes, we use it to improve stance detection. We combine BERTSCORE (DA) with NLI models, initially classifying tweet-misconception pairs with high BERTSCORE scores (>0.4) as Relevant, subsequently using the NLI model to determine whether the pair Agree or Disagree. We denote such "combined models" by inserting a plus sign between the retrieval model and the NLI model, e.g., BERTSCORE (DA) + BiLSTM denotes a model that uses BERTSCORE (DA) to determine retrieve relevant misconceptions and a BiLSTM NLI model for classifying the stance.

**Results** Stance detection results in Table 4 show that, generally, most models do not perform well on the Agree and Disagree classes, which are minority classes in our dataset. On the other hand, performance on No Stance is high; quite a few models achieve an F1-score of 89% or higher. BERTSCORE (DA) + SBERT (DA) (on MultiNLI) achieves the highest F1 (41.2) for the Agree class, while also obtaining the highest macro averaged Precision (55.9%) and F1 (50.2). The combined BERTSCORE (DA) + NLI approach, in general, improves F1 across all classes for all models.

| I | II | III | IV | Example |
|---|----|-----|----|---------|
| ✓ | ✗ | ✗ | ✗ | **Tweet:**In order for accurate information about the #coronavirus to be obtained, you have to be able to do widespread testing. The U.S. is behind many other countries for a variety of reasons. #COVID19 #COVID19US https://t.co/muzxHjY0XP
**Misconception:** Anybody in the U.S. who wants a COVID-19 test can get a test. |
| ✓ | ✓ | ✗ | ✗ | **Tweet:** There is evidence that coronaviruses can live on inanimate surfaces for up to nine days, but its not yet clear how likely humans are to be infected by touching these surfaces. https://t.co/DJ99AAISWw
**Misconception:** Sars-cov 2 can survive for prolonged periods of time on surfaces. |
| ✓ | ✓ | ✓ | ✗ | **Tweet:** Covid-19 is about 43 times more deadly if you get it, but China's number of cases is leveling off at around 80K which is much less than the number of US flu cases. If the number of cases is kept small then Covid-19 will be minor compared to the flu.
**Misconception:** COVID-19 is only as deadly as the seasonal flu. |

Table 5: **Misconception Retrieval Examples**. We present examples to demonstrate the difference in performance between some of the semantic similarity models: **(I)** BERTSCORE (DA), **(II)** Avg. BERT (DA), **(III)** Vanilla BERTScore, and **(IV)** Avg. GloVe. ✓= The model retrieved the relevant misconception for a tweet with rank 1; ✗= The model did not score the relevant misconception for the tweet with rank 1.

| Models | Labels | Example |
|--------|--------|---------|
| SBERT | Disagree | **Tweet:** @IVANISTHEMAN @OriginalDWoods @jboog3000 It didnt come from an animal chinese spies smuggled a form of the virus to china from a canadian lab and was then "leaked". |
| SBERT (DA) | Agree | **Misconception:** Coronavirus was taken from a Canadian lab or is the result of bioweapons defense research in China. |
| BiLSTM | No Stance | **Tweet:** @Acyn The corona virus can live on a surface for up to 9 days. Just saying. |
| BERTSCORE (DA) + BiLSTM | Agree | **Misconception:** Sars-cov 2 can survive for prolonged periods of time on surfaces. |
| SBERT (DA) | No Stance | **Tweet:** @alexsalvinews Alex. Check out Dean Koontz, The Eyes of Darkness. 1981. He predicts the Wuhan-400 virus. He said in "around" 2020, a pneumonia-like virus will be spread worldwide. |
| BERTSCORE (DA) + SBERT (DA) | Agree | **Misconception:** Dean Koontz predicted the pandemic in his 1981 novel The Eyes of Darkness. |

Table 6: **Stance Detection Examples**. Presenting examples of cases where combining or domain adaptation lead to flipping prediction to the correct class (Agree). All models here are trained on MultiNLI.

Examples of stance predictions in Table 6 illustrate the differences between these models. The first example demonstrates that knowledge about the domain vocabulary helps domain adapted models in predicting the correct stance, as it did for retrieval. The remaining two examples both show the advantage of the combined BERTSCORE (DA) + NLI approach, in particular, demonstrating that retrieval models are effective at identifying relevant misconceptions, which the NLI models are then able to correctly classify the stance of.

## 5   Related Work

**COVID-19:** In the social sciences, there have been recent efforts to quantify COVID-19 misinformation on social media (Brennen et al., 2020; Kouzy et al., 2020), as well experimental efforts to prevent propagation of misinformation (Pennycook

et al., 2020). At the same time, members of the NLP community have been working on developing tools for the automatic detection of COVID-19-related misinformation online. Serrano et al. (2020) detect YouTube videos spreading conspiracy theories using features of user comments, and Dharawat et al. (2020) classify tweets by the severity of health risks associated with them. McQuillan et al. (2020) study the behaviour of COVID-19 misinformation networks on Twitter using mapping, topic modeling, bridging centrality, and divergence. Penn Medicine launched a chatbot to provide patients with accurate information about the virus (Volpp-Kevin et al., 2020), and a crowdsourced chatbot, Jennifer, is also available to answer questions about the pandemic (Li et al., 2020). We are the first to frame COVID-19 misinformation detection as a two-stage task of misconception retrieval and pair-

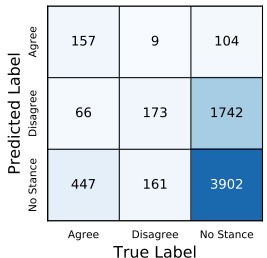 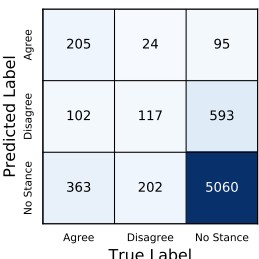

(a) SBERT (DA)   (b) BERTSCORE (DA) + (a)

Figure 3: **Confusion Matrices** for stance detection task using SBERT (DA) models trained on MultiNLI. The second model uses BERTSCORE (DA) to first determine whether a misconception-tweet pair is Relevant or No Stance, and only Relevant pairs are further classified by SBERT (DA).

wise classification of stance, and add to this body of work by providing a dataset and benchmark models for automated identification of misinformation.

**Misinformation Detection:** There are several datasets for misinformation detection with binary veracity labels , for example, FakeNewsNet (Shu et al., 2017, 2019, 2020) consisting of news articles, Some Like It Hoax (Tacchini et al., 2017) consisting of Facebook posts, and PHEME (Zubiaga et al., 2018) containing twitter threads.

Misinformation detection is also closely related to fact-checking since both tasks aim to assess the veracity of claims. FEVER (Thorne et al., 2018, 2019) is a dataset of claims and evidence pairs with Supported, Refuted or NotEnoughInfo labels to facilitate research in automated fact checking. This is similar to Emergent (Ferreira and Vlachos, 2016), a stance classification dataset consisting of rumored claims and associated news articles with labels of For, Against, or Observing the claim. Stance detection is also the focus of the Fake News Challenge (FNC-1)[3] consisting of pairs of news article headlines and body texts with Agrees, Disagrees, Discusses, and Unrelated labels.

Our proposed models for detecting misinformation by using classifiers fall within the framework of detecting misinformation using content features (Volkova et al., 2017; Wei and Wan, 2017). Other approaches include using crowd behaviour (Tschiatschek et al., 2018; Mendoza et al., 2010), reliability of the source (Lumezanu et al., 2012; Li et al., 2015), knowledge graphs (Ciampaglia et al., 2015), or a combination of these approaches

(Castillo et al., 2011; Kumar et al., 2016). Adapting these techniques to COVID-19 misinformation is a promising direction for future work.

## 6   Conclusions and Future Work

The ongoing COVID-19 pandemic has been accompanied by a corresponding 'infodemic' of misinformation about the virus. It is important to develop tools to automatically detect misinformation online, especially on social media sites where the volume and speed of the spread are high. However, rapidly evolving information and novel language make existing misinformation detection datasets and models ineffective for detecting COVID-19 misinformation. In this paper, to initiate research on this important and timely topic, we introduced COVIDLIES, a benchmark dataset containing known COVID-19 misconceptions accompanied with tweets that Agree, Disagree, or express No Stance for each misconception, annotated by experts. Our code, dataset, and a demo of our best performing system are publicly available at https://ucinlp.github.io/covid19.

Given a tweet, we formulate the task of detecting misinformation as retrieving relevant misconceptions, and classifying whether the tweet supports or refutes it. We evaluate a number of approaches for this task, including common semantic similarity models for retrieval, accurate models trained on a variety of NLI datasets, and domain adaptation by pretraining language models on a corpus of COVID-19 tweets. We demonstrate domain adaptation significantly improves results for both subtasks of misinformation detection. We also show that it is feasible to detect the stance of tweets towards misconceptions using models trained on existing NLI datasets. However, the performance has considerable scope for improvement since existing NLI datasets do not contain texts on COVID-19 and are linguistically different from tweets.

Future work will involve using models trained on more domain specific and linguistically similar text. We plan to continually expand our annotated dataset by including posts from other domains such as news articles and Reddit, and misconceptions from sources beyond Wikipedia, such as Poynter (2020). We invite researchers to build COVID-19 misinformation detection systems and evaluate their performance using the presented dataset.

---

[3]http://www.fakenewschallenge.org/

## Acknowledgements

We thank the anonymous reviewers of EMNLP 2020 NLP COVID-19 workshop for their comments and the authors of related work for publishing their code and data. We would like to acknowledge Nicole Woodruff, an undergraduate at UCLA, Aileen Guillen, a medical student at UCI, Sadhika Yamasani, and Victoria Rong, undergraduates at UCI, for volunteering to help annotate data for this project. We would also like to thank Lidia Flores, staff research associate in Dr. Young's Lab, for her help in compiling tweets. Finally, we would also like to thank Elena Kochkina and Maria Liakata of the Alan Turing Institute for helpful discussions and feedback. This material is based upon work sponsored in part by NSF award #IIS-1817183 and in part by the DARPA MCS program under Contract No. N660011924033 with the United States Office Of Naval Research.

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
