# OpenReview forum: "COVIDLies: Detecting COVID-19 Misinformation on Social Media"
_EMNLP/2020/Workshop/NLP-COVID — NLP-COVID19-EMNLP Oral_

### Official Review · AnonReviewer3 · 2020-09-17
**A well written paper with clear contributions and valid experiments**

**Rating:** 9
**Confidence:** 5

**Review:**

This paper purposed a method for COVID-19 related misinformation detection. The authors contribute a datasets with 5K annotated tweets on 86 COVID related misinformation. They evaluated many approaches for misinformation detection and stance detection, and the domain adapted BERTscore  and SBERT achieved the best performance in these two tasks. The paper is written in a clearly and concise way, and well motivated.

Some further work is suggested for the authors to consider. In this study, the misinformation in tweets is annotated by experts. To adapt to new misinformation and automate the process, the authors can link the information in tweets with peer-reviewed scientific publications to detect the new misinformation in tweets.

---

### Official Review · AnonReviewer1 · 2020-09-20
**A dataset and 2-stage pipeline for detecting COVID-19 related misinformation on twitter**

**Rating:** 9
**Confidence:** 4

**Review:**


The authors generate a dataset containing 5000 related to 86 covid-19 misconceptions. They also deploy a two-step process for classifying tweets. At the first step, they deploy a semantic model that tries to define the similarity of a tweet to one of the 86 misconceptions. Then, they perform stance detection to investigate if the tweet agrees to the specific misconception or not. They use different models and techniques coming from NLI to investigate the abilities of models in misinformatio detection

Reasons to accept:
* Authors generate an new benchmark dataset for covid-19 misinformation detection.
* They create a smart two step process for detecting misinformation
* They investigate if general NLI archtiectures that are not domain adapted can help in misinformation detection
* They use different architectures and evaluate their accuracy
* They definitely provide new knowledge and interesting insights on how misinformation can be detected.

Reasons to reject:
* Models accuracy is generally low. That could be due to the limited amount of observation (5000) and the high amount of potential classes (86 * 3)

Comments to the authors:
I enjoyed the study and I definitely support its publication. One thing that I would additionally want to see is comparing your results with these of a standard fine-tuned language model classifier. I would assume yours would be better, but such a comparison would provide clear evidence why shemantic association and stance detection is a better process to follow.

---

### Official Review · AnonReviewer2 · 2020-09-23
**COVID-Lies for measuring COVID-19 misinformation on social media**

**Rating:** 8
**Confidence:** 4

**Review:**

The authors develop a large dateset of 86 misconceptions along with 5000 annotated tweet-misconception pairs from COVID-19 related tweets that were generated in March and April of 2020. All tweets were manually labeled by researchers from the UCI School of Medicine as: 1) having a positive expression of the misconception, 2) contradicting/disagreeing with the misconception or 3) being neutral or not relevant to the misconception. The researchers then evaluate the performance of the existing NLP systems on the dataset.

While the authors provide a good framework for developing and testing misinformation in social media, given the rapidly evolving/changing evidence base related to COVID-19, strong consideration should further be given to developing more automated systems for classifying social media content (e.g. bench-marking them against the content on the reputable websites, such as CDC, WHO or PHAC, rather than relying on human annotators to maintain the misconception databases current). In the meantime, COVID-Lies provides a great starting point for the development of robust misinformation detection systems and their evaluation.